# Virulence Pattern Analysis of Three *Listeria monocytogenes* Lineage I Epidemic Strains with Distinct Outbreak Histories

**DOI:** 10.3390/microorganisms9081745

**Published:** 2021-08-16

**Authors:** Martin Wagner, Jörg Slaghuis, Werner Göbel, José Antonio Vázquez-Boland, Kathrin Rychli, Stephan Schmitz-Esser

**Affiliations:** 1Unit for Food Microbiology, Institute for Food Safety, Food Technology and Veterinary Public Health, University of Veterinary Medicine, 1210 Vienna, Austria; Kathrin.rychli@vetmeduni.ac.at; 2Austrian Competence Center for Feed and Food Quality, Safety and Innovation, 3430 Tulln, Austria; 3Microbiological Analytics, Merck KGaA, Frankfurterstrasse 250, 64293 Darmstadt, Germany; joerg.slaghuis@merckgroup.com; 4Cair of Microbiology, University of Würzburg, Am Hubland 1, 97074 Würzburg, Germany; goebel@biozentrum.uni-wuerzburg.de; 5Microbial Pathogenesis Laboratory, Edinburgh Medical School (Biomedical Sciences-Infection Medicine), University of Edinburgh, Edinburgh EH16 4SB, UK; v.boland@ed.ac.uk; 6Department of Animal Science, Iowa State University, Ames, IA 50011, USA; sse@iastate.edu

**Keywords:** pathogenicity, whole-genome analysis, prolonged survival

## Abstract

Strains of the food-borne pathogen *Listeria* (*L*.) *monocytogenes* have diverse virulence potential. This study focused on the virulence of three outbreak strains: the CC1 strain PF49 (serovar 4b) from a cheese-associated outbreak in Switzerland, the clinical CC2 strain F80594 (serovar 4b), and strain G6006 (CC3, serovar 1/2a), responsible for a large gastroenteritis outbreak in the USA due to chocolate milk. We analysed the genomes and characterized the virulence in vitro and in vivo. Whole-genome sequencing revealed a high conservation of the major virulence genes. Minor deviations of the gene contents were found in the autolysins Ami, Auto, and IspC. Moreover, different ActA variants were present. Strain PF49 and F80594 showed prolonged survival in the liver of infected mice. Invasion and intracellular proliferation were similar for all strains, but the CC1 and CC2 strains showed increased spreading in intestinal epithelial Caco2 cells compared to strain G6006. Overall, this study revealed long-term survival of serovar 4b strains F80594 and PF49 in the liver of mice. Future work will be needed to determine the genes and molecular mechanism behind the long-term survival of *L. monocytogenes* strains in organs.

## 1. Introduction

The Gram-positive bacterium *Listeria monocytogenes* is the causative agent of listeriosis, a rare, but severe food-borne infection in humans and animals. In healthy individuals, listeriosis is usually restricted to a non-invasive self-limited gastroenteritis, which typically occurs 24 h after ingestion of a large inoculum of bacteria and usually lasts two days [1]. In immunocompromised individuals and elderly, an invasive and systemic infection can occur within an incubation time of 20 to 30 days with a high mortality rate of 25–30%. Furthermore, pregnant women and neonates are a high-risk group. Invasive *L. monocytogenes* infection of non-pregnant individuals typically manifests as bacteremia/septicemia or as central nervous system infection including meningitis, meningoencephalitis and with lower prevalence also rhombencephalitis and brain abscess [2,3,4]. In pregnant women, *L. monocytogenes* can infect the fetus or unborn, leading to spontaneous abortion or stillbirth. Neonatal listeriosis may represent as sepsis or meningitis with severe consequences and a high case fatality of 20% [5].

There is evidence for the difference in the virulence and pathogenic potential of *L. monocytogenes* strains [6,7,8]. *L. monocytogenes* strains can be grouped into four major phylogenetic lineages (I, II, III and IV), and most clinical strains are of phylogenetic lineage I and II [9]. Of the 13 lineage-related serovars only three, namely 1/2a within lineage II, and 1/2b and 4b within lineage I, are responsible for the majority of the clinical cases of human (and animal) listeriosis [10]. Especially, serovar 4b strains cause over 50% of listeriosis cases worldwide in spite of being much less frequently found in food samples as compared to strains from serovar 1/2a and b [6,10]. Furthermore, *L. monocytogenes* strains can be assigned to multi-locus sequence types, which are grouped into clonal complexes (CC). Multi-locus sequence typing is based on the sequences of seven housekeeping genes. Recent advances in *Listeria* infection biology have shown that certain CCs such as CC1, CC2, CC4, and CC6 are more frequently associated with clinical cases, whereas others like CC9 and CC121 are mainly of food-borne origin [6,7,8,11].

The understanding of the pathophysiology of *Listeria* infection has made major steps forward, but there are still new insights to come. After ingestion and survival in the stomach, *L. monocytogenes* traverses the intestinal epithelial barriers via goblet cells and enterocytes into the lamina propria and spreads into the bloodstream through the lymph nodes to disseminate to the target organs of replication, the liver and spleen [2,12]. In case *L. monocytogenes* trespasses the liver barrier, the bacteria shuttle to secondary sites of clinical manifestation, the placenta, and the central nervous system. An arsenal of virulence factors mediates the complex pathogenicity of *L. monocytogenes* among them the internalins InlA and InlB, which are required for the entry into non-phagocytic cells [12,13,14]. Subsequent to the invasion step, *L. monocytogenes* is temporarily enclosed in a phagocytic vacuole, before it escapes into the host cell cytoplasm, which is mediated by the listeriolysin O (LLO) and the phospolipases (PlC) A and B. Once inside the host cell cytosol, *L*. *monocytogenes* can survive, replicate and move to the neighboring cell. The actin-assembly-inducing protein ActA is essential for intracellular motility and cell-to-cell spread. Additionally, there are numerous other virulence factors that e.g., hijack cellular processes, modulate the immune system reach even the eukaryotic nucleus and influence gene expression [12].

One puzzling issue regarding the pathogenicity of *L*. *monocytogenes* is the long incubation period of the invasive manifestation in humans and animals, varying from 24 h in cases showing gastroenteritis to 9 in cases with central nervous system infection and 27.5 days in cases with abortions, respectively [15]. This unusually long incubation period required for the manifestation of clinical systemic listeriosis implies that *L. monocytogenes* might persist in host tissues for extended periods without causing overt clinical signs. In a recently published paper, Vazquez-Boland and co-workers stipulated the hypothesis that hypervirulent strains, which survive longer in the liver and the spleen of the host, have a higher probability of secondary dissemination of the brain and the placenta [16]. Two hypervirulent 4b strains P14 and PF49, both assigned to CC1, were still detectable in the mouse liver three weeks post infection [16]. P14 was isolated from an adult patient with central nervous symptoms in Spain, and PF49 is an epidemic strain from a cheese-associated outbreak in Switzerland [17]. This observation is striking since most mouse studies have shown that the infection is cleared from the liver seven days post infection, when a sublethal dose of *L. monocytogenes* is administrated.

In this work, we follow-up this hypothesis using the serovar 4b strain PF49 (CC1) for an in-depth characterisation of its virulence properties. PF49 was associated with a large community-acquired epidemic of invasive listeriosis with a high proportion of brain infections (79%) [17]. We compared the genome of strain PF49 with strains F80594 (CC2) and G6006 (CC3). Strain F80594, also serovar 4b, has caused gastrointestinal listeriosis in a mainly healthy population [18] and strain G6006 (serovar 1/2b) was isolated during a large outbreak of gastroenteric listeriosis caused by the consumption of a heavily contaminated milk product [19] (Table 1). We characterized the virulence potential of these listeriosis strains in intestinal epithelial, hepatocytic, and phagocytic cells. We aimed to mimic the natural route of infection (translocation across the epithelium, infection of the liver, and contact with phagocytes in the bloodstream) and determined the pathogenicity in a mouse infection model. The laboratory isolate EGD was included as a control.

## 2. Materials and Methods

### 2.1. Bacterial Strains

The strains were stored at −80 °C in 20% glycerol phosphate-buffered saline, pH 7.4 (PBS) and grown in brain-heart infusion (BHI, Merck, Darmstadt, Germany) broth or on agar at 37 °C. Inocula for in-vitro and in-vivo studies were prepared from early stationary phase cultures. Bacteria were harvested by centrifugation, washed in PBS, resuspended in glycerol-PBS and stored at −80 °C in 500 µL aliquots. The exact number of bacteria in the frozen aliquots was calculated by plate counting.

### 2.2. DNA Isolation, Genome Sequencing, and Analysis

The *L. monocytogenes* strains were cultivated under aerobic conditions at 37 °C in BHI with 125 rpm shaking and harvested by centrifugation. The resulting pellet was used for DNA isolation using the QIAGEN genomic-tip columns and buffers (QIAGEN), following the recommendations of the manufacturer. For G6006 and PF49, genome sequencing was performed using an Illumina GAII genome analyzer at the University of Veterinary Medicine Vienna, Austria. Sequencing was performed using paired-end sequencing technology and 100 bp read-length using Illumina standard protocols. For F80594, genome sequencing was performed using an Illumina HiSeq2000 sequencer at the Campus Science Support Facilities (CSF) Next-Generation Sequencing unit, Vienna, Austria, using paired-end sequencing technology and 100 bp read length. Three (F80594) and ten (G6006, PF49) million reads were used for a *de novo* assembly using ABySS [20] with a k-mer size of 64. For G6006, the assembly resulted in 28 contigs (>500 bp) with an average coverage of 319×. For PF49, the assembly resulted in 26 contigs (>500 bp), for which the average coverage was 330×. For F80594, the assembly resulted in 24 contigs with an average coverage of 114×. The draft genome sequences of the *L. monocytogenes* strains were annotated and analyzed using PATRIC, www.patricbrc.org (accessed on 15 June 2021) [21]. The presence and absence of key virulence genes was determined using BlastP searches using a list of 95 virulence proteins as recently described [22].

### 2.3. Mouse Infection Assays

Groups of female BALB/C mice weighing 18 to 22 g (six per strain and time point, Charles River) were infected intravenously (iv) with a sublethal dose (5 × 10^3^ CFU) of *L. monocytogenes* and the bacterial load in the liver was monitored at time points 3, 5, 7, 10, 12, 14 and 20 days post infection (p.i.). At the specified time points, the mice were humanely sacrificed by exposing them to carbon dioxide. The liver was aseptically removed and homogenised in 10 mL PBS, and the number of CFU per organ was determined by standard plate counting. The mean 50% lethal doses (LD_50_) were tested using groups of five BALB/C mice, which were injected with 200 μL suspensions containing 10^3^, 10^4^, 10^5^, 10^6^, and 10^7^ bacteria. The survival was monitored during a 7-day period. The control group was injected with 200 μL sterile PBS; LD_50_ values were determined according to the method of Reed and Muench [23].

### 2.4. In-Vitro Virulence Assays

Invasiveness, intracellular proliferation, and cell-to-cell spread were determined using human intestinal epithelial Caco-2, the mouse macrophage-like P388D1, and the mouse liver TIB-73 cell lines. Cells were cultivated in 24-well dishes (Greiner, Frickenhausen, Germany) in 1 mL of RPMI 1640 medium supplemented with 2 mM L-glutamine and 10% heat-inactivated foetal calf (FCS) serum under 5% CO_2_–95% O_2_ atmosphere at 37 °C. Cells were seeded in RPMI-complete medium 72 h prior to infection. The medium was changed 24 h before infection. A gentamicin protection assay was used to determine the invasiveness. Semiconfluent (80%) cell monolayers were infected with a *L. monocytogenes* suspension in RPMI at a multiplicity of infection of 20 for 45 min (P388D1 macrophages) and 60 min (Caco-2 and TIB-73 cells), washed three times with PBS and incubated for 30 min with fresh RPMI medium supplemented with 25 µg/mL gentamicin (Serva, Heidelberg, Germany) to kill extracellular bacteria. The cells were washed, lysed by the addition of 1 mL ice-cold distilled water followed by incubation on ice for 15 min, and the intracellular bacterial numbers were determined by plate counting using tryptic soy agar. The intracellular proliferation was determined by calculating the intracellular growth coefficient with the formula IGC = (CFU_6h_ − CFU_invasion_)/CFU_invasion_. To measure the cell-to-cell spread, cells were infected with strains harbouring the plasmid pLSV16*gfp-a*. Transformation of *L. monocytogenes* PF49, F80594, G6006, and EGD was performed as described earlier [24]. The plasmid stability was confirmed for at least 36 h in cell culture medium supplemented with 7.5 µg tetracycline. Infection was performed as described above and the number of intracellular bacteria was counted after 6 h. For flow cytometry analysis, Caco-2 cells were detached from the dishes by incubation with 200 µL of trypsin solution (Trypsin-EDTA; Gibco Invitrogen Life Tech., Paisley, UK) for 5 min. The cells were then suspended in 1 mL PBS and transferred into plastic tubes. GFP fluorescence emitted by intracellular bacteria was measured by flow cytometry using an Epics Elite ESP cell sorter (Coulter, Krefeld, Germany) and the percentage of positively stained host cells was determined (488 nm line of an argon ion laser, green light channel: 525/10 nm bandpass filter, counts: 10,000 cells per sample per run). All experiments were repeated three times.

### 2.5. Statistics

Statistical analysis was performed using Excel16.0 and SPSS.24 software. The mean values and standard deviations (SD) were calculated. The Welch-test was used to determine variance homogeneity. The Tukey-HSD post hoc test (variance homogeneity) was used to determine significant differences between the IGC and the percentage of GFP positive cells of the different strains and the number of bacterial load in the liver at the different time points. *p* < 0.05 was considered as a statistically significant difference.

## 3. Results

### 3.1. Whole Genome Sequencing

For the 1983–1987 Swiss Vacherin cheese outbreak, the source of strain PF49, one additional complete genome was available (strain LL195) [25]. Additional to G6006, two genomes were available from strains of the milk outbreak in Illinois: FSL-R2-503 (G6054) and FSL-R2-502 (Table 2). The genome sequences of all listeriosis outbreak isolates revealed typical features of L. monocytogenes genomes. A general overview of the genomic properties and features is provided in Table 2. None of the strains contained a plasmid except for strain FSL-F2-502, which harboured a 57 kbp plasmid. The strains harboured between one and three prophages.

The Blast search using 95 *L. monocytogenes* virulence factors as query sequences, revealed a high conservation of the majority of virulence factors (same length and an amino acid identity higher than 90%). Among the major virulence factors, we observed only differences in the ActA sequence. F80594, G6006, FSL-R2-503 and FSL-R2-502 harboured an ActA4 variant which is comprised of 604 amino acids, whereas PF49 and LL195 had an ActA3 variant with 604 amino acids [26] (Table 3). Interestingly, PF49 harboured a truncated Vip protein (Lmo0320) resulting in 335 amino acids compared to 399 amino acids in EGD-e. Vip in the other strains had a length of 418 amino acids. G6006 and the other outbreak strains harboured the *aut* gene, encoding Auto, an autolysin which is important for *L. monocytogenes* virulence [27]. In contrast, Auto is absent from F80594 and PF49, which instead harbored IspC, another autolysin primarily found in serovar 4b strains [28]. Another notable difference between the strains studied here were different variants of Ami, an additional autolysin encoded by the *lmo2588* gene [29,30]. G6006 encodes a full-length homolog of the EGD-e Ami consisting of 917 amino acids and sharing 99% amino acid identity, whereas F80954 and PF49 encoded a shorter version of Ami with 770 amino acids with 72% amino acid identity to the EGD-e Ami. The shorter Ami variants are due to a lower number of C-terminal GW repeats. Moreover, all strains harboured an Internalin J (InlJ1) variant, and the listeriolysin S locus was present in all strains except strain F80594.

The additional outbreak strain genomes showed the same genetic features as the isolates included in the in-vitro and in-vivo assays with a few exceptions: there are possible truncations in InlJ and GtcA in LL195 compared to PF49 and a 57 kbp plasmid is only present in FSL-F2-502.

### 3.2. Animal Behaviour, Recovery of L. monocytogenes from the Liver and LD_50_

The behaviour of the mice was monitored visually daily post infection. Mice of groups infected with strain PF49, F80594, and EGD became apathetic three to five days after infection, displaying shaggy fur and reduced mobility. Upon day 5, all mice infected with these L. monocytogenes strains returned to normal behaviour. In contrast to the other strains, the mice infected with strain G6006 remained healthy throughout the observation period.

The two 4b strains, PF49 and F80594, showed different bacterial load patterns in the liver than strain G6006 and EGD (Figure 1, Appendix A).

On day three (acute infection phase), infecting mice with strain PF49 resulted in a significantly higher bacterial load in the liver than with strain G6006 and EGD. On day 5, the bacterial load of mice infected with strain G6006 was significantly lower compared to the other three strains. PF49 and F80594 were still recovered at day 5 from the liver of all six animals in high numbers, whereas strain G6006 was only recovered from the livers of two animals in low numbers. At subsequent time points, strain G6006 was no longer detectable in the liver. Strain EGD was found at day 7 in the liver of six animals in low numbers. At later time points, EGD was not recovered from the liver. In contrast, strains PF49 and F80594 were still detectable in the liver up to day 20 and 14 p.i., respectively. These results demonstrate that PF49 and F80594 have an enhanced capacity to survive in vivo in the liver. Furthermore, the LD_50_ of strain G6006 and EGD (log CFU 5.83 ± 0 and 5.07 ± 0.3) was higher compared to strain PF49 and F80594 (log CFU 4.620 ± 0 and 4.66 ± 0.2).

### 3.3. Invasiveness, Intracellular Proliferation, and Spreading in Host Cells

To get insights into the invasiveness, intracellular proliferation, and spreading, we performed in-vitro assays.

No difference in the invasion efficiency of all four strains (PF49, F80594, G6006, and EGD) was observed in Caco-2 and TIB-73 cells. Furthermore, the uptake of all strains by macrophage-like P388-D1 cells was equal (data not shown). In Caco-2 a higher intracellular proliferation (calculated as IGC) of strain G6006 and EGD was observed compared to strain PF49 and F80594; however, the effect was not statistically significant (Figure 2A). In contrast, a significantly higher number of GFP-positive Caco-2 cells were observed after infection with strain PF49 and F80594 (37.1 and 23.6%) compared to strain G6006 and EGD (7.5 and 3.8%, Figure 2B). In TIB-73 cells, no differences between the strains were observed for intracellular growth and cell-to-cell spread (Figure 2C,D). In macrophage-like P388D1 cells, the IGC was lower for strain G6006 and EGD than for strain PF49 and F80594. However, the differences were only small and not significant. In contrast, an increased cell-to-cell spread was observed for strain EGD compared to the epidemic strains (Figure 2E,F).

## 4. Discussion

In this study, we combine genome analysis with in vitro virulence assays and mouse infection experiments for up to 20 days to unravel the pathogenicity of outbreak-associated *L. monocytogenes* strains. This study is thought to create some hypotheses why lineage 1 outbreak clones of serovar 4b might have led to a clinical outcome that was obviously very different from outbreaks of non-serovar 4b isolates. It further provides a hypothesis why some *Listeria monocytogenes* strains tend to persist in the murine body what was proven for PF49 and P14 [16].

The CC1 strain PF49 is an interesting isolate. Although the estimated infective dose was lower in the outbreak associated with PF49 than in the outbreak caused by G6006 (log CFU of 6 for strain PF49 and log CFU of 11 for strain G6006), the Swiss outbreak was remarkably severe [17,19]. A high percentage of meningoencephalitis cases was also recorded in the group of non-immunocompromised individuals what was not explainable [17]. The low LD_50_ value and the virulence properties proved PF49 to be a true hypervirulent isolate. This is in accordance with the findings of other studies that CC1 is overrepresented among clinical and epidemic strains, often isolated from patients with fever and no immunocompromising comorbidities [6,7,8]. Moreover, CC1 strains seem to be strongly associated with dairy products [31], which is consistent with the finding that soft cheese was the source of PF49. In contrast, the 4b strain F80594 was isolated from a small outbreak in a day-care facility in Denmark, in which three children (all of age two) were involved. The clinical manifestations were vomiting and fever. *L. monocytogenes* could be isolated from the blood of one patient [18]. The outcomes of the epidemiological follow-up investigations were limited. Neither an estimate of the ingested dose nor the exact food source are available. The isolate F80594 was assigned to CC2. Strains of CC2 were, like CC1 strains, more frequently isolated from clinical than from food-associated sources [6,11,32]. Moreover, CC2 strains are more frequently isolated from dairy than from meat and fish products [11,31]. The 1/2b serovar strain G6006 is assigned to CC3, an intermediate CC showing equal distribution among clinical and food isolates [6,31]. G6006 was responsible for a large outbreak of febrile gastroenteritis due to contaminated chocolate milk in the United States, including only one case with central nervous symptoms [19]. CC3 does not seem to be associated with a specific food type [11,31].

We detected differences in the colonization and clearance of the liver, and in the spreading capacity. In the mouse experiment, we detected that strains PF49 and F80494 showed significantly different colonization and clearance of the liver compared to strain G6006 and strain EGD. In line with the different clinical manifestations, the highest bacterial load was found in mice infected with strain PF49, followed by strain F80494 [17,18]. Genetic differences in virulence-associated genes between the two 4b strains, with an average nucleotide identity of 99.6%, were rare. PF49 harboured a different ActA variant than strains F80594 and G6006. F80594 and G6006 had an ActA4 variant, which is slightly longer than the ActA variant of strain PF49 and EGDe [26]. ActA is responsible for bacterial movement and cell-to-cell spread. Moreover, it has recently been shown to be also involved in aggregation as well as long-term colonization of the gut [33,34]. The effect of the different ActA variants on the virulence potential and pathogenicity of *L. monocytogenes* is not well understood. We can only speculate that the ActA3 variant of strain PF49 might cause the higher bacterial load at day three due to longer colonization of the gut.

Strains G6006 and EGD revealed lower bacterial loads in the liver at day three post infection, further decreasing at the following time points. This is the expected behaviour of *L. monocytogenes* in the liver of intravenously infected naïve wild-type mice. Clearance of strain G6006 was faster than that of strain EGD. On day seven, no bacteria were detectable in the liver. The decrease in the bacterial numbers is a consequence of an effective macrophage activation and protective immune response mediated by Th1 and CD8^+^ T-cells [35,36]. In contrast, bacteria were recovered from the liver of mice infected with the two 4b strains, up to day 14 for F80494 and day 20 for PF49, respectively. We speculate that some clones might escape the immune defence, thus leading to a more deferred style of infection. In addition to PF49, this phenomenon was found also in the CC1 strain P14, isolated from a patient with CNS manifestation during an outbreak in Spain [37]. Both 4b strains showed higher spreading in human intestinal epithelial Caco2 cells compared to strain G6066 and the lab strain EGD. However, these effects were only minor. Spreading in the mouse liver TIB-37 and macrophage-like PD388D1 cells was similar between the three outbreak strains. Only strain EGD showed significantly higher spreading capacities in the macrophage-like cells. No differences in the invasion efficiency, uptake by macrophage-like cells, and intracellular growth were observed.

For the sequence analysis, we have used a set of 95 *L. monocytogenes* virulence genes as query sequences for BlastP to identify virulence gene differences among the outbreak clones. As almost all of the major virulence genes were found highly similar to the reference sequences (>90% amino acid identity), we infer that the observed differences in virulence might be either due to posttranscriptional effects or unstudied metabolic pathways. However, some small differences in the virulence associated genes were observed. Notable is the presence/absence of the autolysins: IspC, known to be important for virulence [28], is only present in the two 4b strains, whereas G6006 harboured the autolysin Auto (Lmo1076). It has been speculated that IspC could be responsible for crossing the blood-brain barrier [28], which might contribute to the high percentage of central nervous system complications in the Swiss cheese outbreak. It is currently unclear if the presence of IspC might result in higher virulence in 4b strains, since Auto also has a role in virulence in *L. monocytogenes* [27]. Another genomic feature that might explain the difference in virulence in mice between the strains might be differences in the autolysin Ami, which is important for adhesion and internalization [29,30]. G6006 harbours an Ami homolog with high similarity (99% amino acid identity) and the same length (917 amino acids) as the functionally characterized EGD-e Ami. Interestingly, the Ami homologs in F80594 and PF49 are shorter (770 amino acids) and show lower amino acid identity (72%) to the EGD-e Ami. The lab strain EGD showed comparable virulence to G6006 in mice and similar spreading in Caco2 cells and Ami and Auto were most similar to that of G6006. Future studies are needed to identify the function and the virulence potential of the different autolysin homologs.

In this study, we showed that not only strains of CC1 [16], but also of CC2 have the potential to survive for extended time periods in the liver, enhancing the possibility for dissemination to the secondary organs. Experiments, e.g., transcriptome analysis of bacteria isolated from organs such as the liver, are needed to determine the molecular mechanism behind the long-term survival of some *L. monocytogenes* strains in this tissue. Additional studies including both a higher number of highly prevalent CCs overrepresented by clinical and epidemic strains and more strains per CC will be necessary to prove whether the higher pathogenicity and severe clinical manifestation is due to a prolonged survival capacity in the intestine, lymph nodes, liver, and spleen.

## Figures and Tables

**Figure 1 microorganisms-09-01745-f001:**
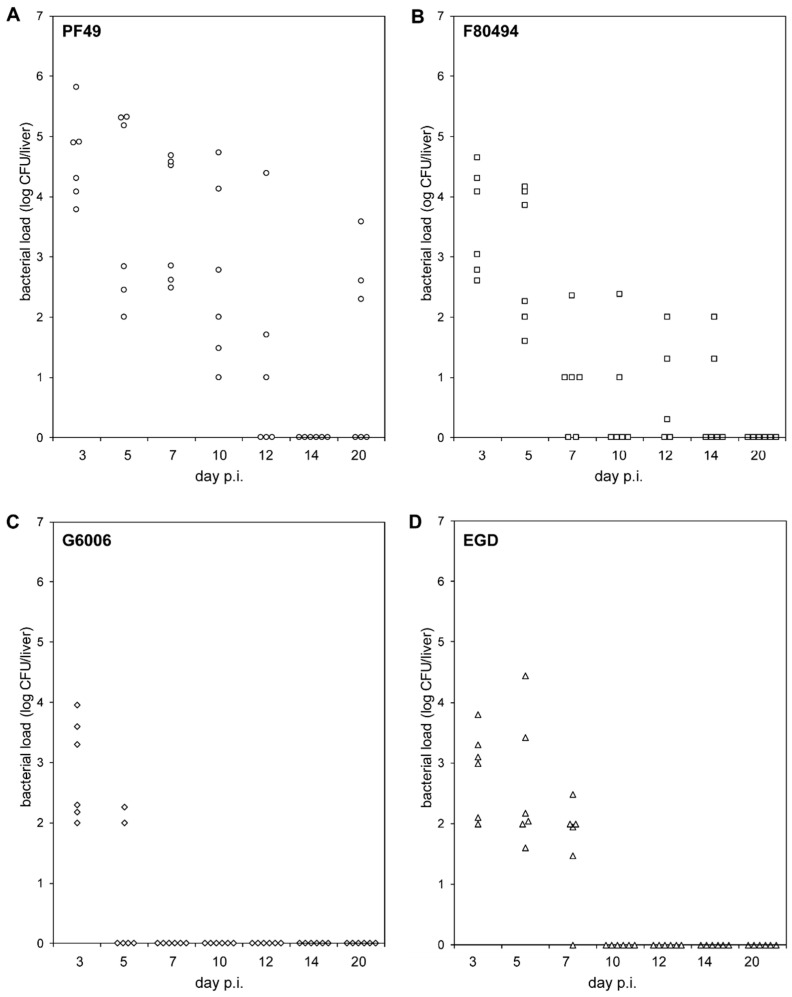
In-vivo pathogenicity. Bacterial loads (log CFU) in the livers of BALB/C mice (*n* = 6) intravenously infected with strain PF49 ((**A**), circle), F80594 ((**B**), square), G6066 ((**C**), rhombus) and EGD ((**D**), triangle) 3, 5, 7, 10, 12, 14 and 20 days post infection (p.i.).

**Figure 2 microorganisms-09-01745-f002:**
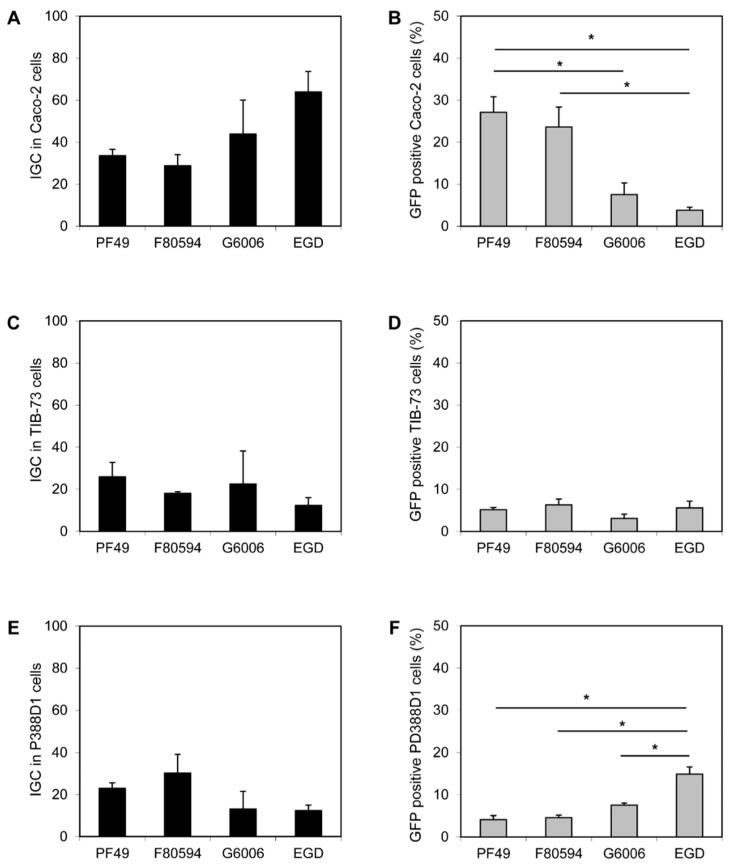
In vitro virulence. Intracellular growth coefficient (IGC) in human intestinal epithelial Caco-2 (**A**), mouse liver TIB-73 (**C**) and mouse macrophage-like PD388D1 cells (**E**) and cell-to-cell spread, shown as percentage (%) of GFP positive Caco-2 (**B**), TIB-73 (**D**), PD388D1 cells (**F**). * statistically significant difference (*p* < 0.05).

**Table 1 microorganisms-09-01745-t001:** Strain information and epidemiological data.

Strain	PF49	F80594	G6006
Source	Cheese	Stool	Milk
Servoar	4b	4b	1/2b
CC (ST)	1 (1)	2 (2)	3 (3)
Origin	Canton of Vaud, Switzerland	Næstved area, Denmark	Illinois, USA
Isolation Year	1986	1995	1994
No. of Cases	122	3	45
Epidemiologic Clone Identified (%)	75	100	92.9
Pregnancy-Associated Cases (%) ^a^	53.3	0	2.2
Non-Pregnant Cases (%)	46.7	100	97.8
Bacteremia (%)	21	33	6.7 ^b^
Central Nerval Symptoms Developed (%)	79 ^c^	0	2.2 ^b^
Diarrhoea (%)	3	100	79
Underlying Diseases (%)	42 ^d^	0	3.3
Infective Dose Estimated (logCFU)	6	Unknown	11
Median of Incubation (days)	Unknown	1	<1
Reference	[17]	[18]	[19]

^a^ Neonate (early onset)/mother cases count as one case. ^b^ Cases revealed by post-epidemic surveillance. ^c^ In the group of adults (*n* = 57). ^d^ Elderly aged >65 years not included.

**Table 2 microorganisms-09-01745-t002:** General genomic features of *L. monocytogenes* outbreak strains.

Strain	PF49	LL195 *	F80594	G6006	FSL-R2-503 * (G6054)	FSL-R2-502 *
Outbreak	Switzserland	Switzserland	Denmark	Denmark	Illinois, USA	Illinois, USA
Source	cheese	human	fecal	milk	human	food
Year	1986	1987	1995	1994	1994	1994
GenBank Accession No.	JADXDN000000000	HF558398	JADXDP000000000	JADXDO000000000	AARR00000000	CP006594; CP006595
Genome Assembly Size	2.918 MKbp	2.904 Mbp	2.991 Mbp	2.988 Mbp	2.991 Mbp	3.034 Mbp
No. Contigs	26	1	24	28	55	2
Plasmid Presence (size)	no	no	no	no	no	yes (57,557 bp)

* Only included in the genome analysis.

**Table 3 microorganisms-09-01745-t003:** Virulence-related genomic features of *L. monocytogenes* outbreak strains PF49, F80594 and G6006 included in this study. Other sequenced outbreak related strains (for the Swiss outbreak: LL195; the US outbreak: FSL-R2_503, FSL-R2-502) were analysed in addition. Amino acid (AA) identities are only shown when lower than 90%.

Protein (Locus_tag)	PF49	LL195 *	F80594	G6006	FSL-R2-503 (G6054) *	FSL-R2-502 *
ActA (Lmp0204)	ActA3 (604 AA)	ActA3 (604 AA)	ActA4 (639 AA)	ActA4 (639 AA)	ActA4 (639 AA)	ActA4 (639 AA)
Autolysin amidase Ami, (Lmo2558)	shorter (605 AA) 75% AA identity	shorter (770 AA) 72% AA identity	shorter (770 AA) 72% AA identity	present (917 AA)	absent	present (917 AA)
Autolysin Auto (Lmo1076)	absent	absent	absent	present	present	present
Autolysin IspC (LMOf2365_1093)	present	present	present	absent	absent	absent
Bacteriocin (Lmo2776)	present	present	present	present	present	present
InlJ (Lmo2821)	InlJ1 variant (916 AA)	Putative pseudogene	InlJ1 variant (916 AA)	InlJ1 variant (916 AA)	InlJ1 variant (916 AA)	InlJ1 variant (916 AA)
Listerolysin S locus (LIPI-3)	present	present	absent	present	present	present
Vip (Lmo0320)	truncated (335 AA)	present (418 AA)	present (418 AA)	present (418 AA)	present (418 AA)	present (418 AA)

* Only included in genome analysis.

## Data Availability

The genome sequences of the three strains sequenced in this study have been deposited in DDBJ/EMBL/GenBank under BioProject accession number PRJNA681816.

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
