# Peer review of "Virulence Pattern Analysis of Three *Listeria monocytogenes* Lineage I Epidemic Strains with Distinct Outbreak Histories"

_microorganisms, 2021, doi:10.3390/microorganisms9081745_

Round 1
Reviewer 1 Report
The manuscript by Wagner and coworkers addresses the issue of possible relationships between genomic differences and virulence of different L.m strains. The authors combine genome sequencing and in vivo infection analyses.
The topic is very interesting, but the manuscript presents flaws, lacks some information, and, overall, needs several improvements (including English and gene nomenclature).
Details referring to the results should be omitted from the abstract.
The introduction is very "essential". A more extensive presentation of the pathogen, its biological features, and the major virulence mechanisms and genes involved, as well a brief explanation of the concept of "clonal complexes" would be welcome and very beneficial for those readers less familiar with Lm.
Results from genome sequencing are very limited and poorly discussed. Moreover, the authors omitted to cite and refer to some major Lm virulence genes (one among others, hlyA, listeriolysin).
Moreover, no experimental data, but only general hypotheses, are provided about the possible relationships between the genomic/genetic differences the authors have focused on in the text, and the virulence features observed in experiments.
Author Response
The manuscript by Wagner and coworkers addresses the issue of possible relationships between genomic differences and virulence of different L.m strains. The authors combine genome sequencing and in vivo infection analyses.
The topic is very interesting, but the manuscript presents flaws, lacks some information, and, overall, needs several improvements (including English and gene nomenclature).
We have carefully revised the manuscript. The manuscript was additionally revised by a native speaker.
Details referring to the results should be omitted from the abstract.
The introduction is very "essential". A more extensive presentation of the pathogen, its biological features, and the major virulence mechanisms and genes involved, as well a brief explanation of the concept of "clonal complexes" would be welcome and very beneficial for those readers less familiar with Lm.
We have now extended the introduction as requested including an explanation of the concept of CCs.
Results from genome sequencing are very limited and poorly discussed. Moreover, the authors omitted to cite and refer to some major Lm virulence genes (one among others, hlyA, listeriolysin).
We apologize for the confusion regarding the results and discussion of the genome sequencing analyses. As described in the methods section, we have used a set of 95 L. monocytogenes virulence genes as query sequences for BlastP to identify virulence genes and their conservation. As almost all of the virulence genes are highly similar to the reference sequences (>90% amino acid identity), we reported only results for virulence genes that showed different presence/absence patterns, different length compared to the described references, or lower (i.e., <90%) amino acid identity. Given the high conservation of the major virulence genes in L. monocytogenes, our analyses aimed to describe the differences between the strains and differences in virulence genes with regards to the described references to possibly identify genomic explanations for the observed differences in virulence.
Moreover, no experimental data, but only general hypotheses, are provided about the possible relationships between the genomic/genetic differences the authors have focused on in the text, and the virulence features observed in experiments.
We are aware that the results of this study are limited. Genome analysis revealed only minor differences between the strains showing long-term survival in the liver and the strains cleared after a few days. Additional experiments e.g. transcriptome analysis of bacteria isolated from organs, are needed to determine the molecular mechanism behind the long-term survival of L. monocytogenes strains in organs.
Reviewer 2 Report
Dear authors,
this paper is interesting and of high level. It is simple, but with a rigorous research plan that make results obtained solid. This paper needs dissemination in the scientific community and could serve as metadata reservoir for international authorities.
I suggest accepting it after minor revisions, which I have reported below.
For example:
Keywords: avoid repeating words already present in the title
Abstract: the text could be improved, being less narrative and including a final consideration.
Typos on numbers and on unit should be double check and harmonized according to the standard.
For example: -80°C or 37°C should be written -80 °C or 37 °C
In vitro must be italicized
Statistical p should be reported as p or P
- monocytogenes should be italicized
Table 2 and table 3 could be made better and clearer
Expression styles like “(log 6 versus log 11 CFU)” should be avoided
Figure 1: Y axis needs a definition and a unit reported in brackets, moreover, Log CFU is not correct and is better to report (Log10 CFU/mL) or (Log10 CFU/g)
Figures 2A, 2C, 2E: See the comment above. The plots has no statistical p value
Figure 2B, 2D, 2F, still on the title of y axis report the definition and then (%)
Discussion is professionally written, although some considerations on the potential of this study should be included. What this study could give, which is the novel contribution on knowledge that this paper could fill? To whom is targeted?
References section contains many typos:
to >i>Listeria monocytogenes
Nat. Rev. Immunol. 2004.
Appl Env. Microbiol

Author Response
Keywords: avoid repeating words already present in the title- changed
Abstract: the text could be improved, being less narrative and including a final consideration. We have revised the abstract.
Typos on numbers and on unit should be double check and harmonized according to the standard.
For example: -80°C or 37°C should be written -80 °C or 37 °C
In vitro must be italicized
Statistical p should be reported as p or P
- monocytogenes should be italicized
We have checked the manuscript and changed it accordingly.
Table 2 and table 3 could be made better and clearer
We clarified Table 2 and 3.
Expression styles like “(log 6 versus log 11 CFU)” should be avoided – deleted
Figure 1: Y axis needs a definition and a unit reported in brackets, moreover, Log CFU is not correct and is better to report (Log10 CFU/mL) or (Log10 CFU/g)
In Figure 1 we depicture the bacterial load as log CFU in the liver. We have changed the definition as bacterial load (log CFU/liver).
Figures 2A, 2C, 2E: See the comment above. The plots has no statistical p-value
Figure 2B, 2D, 2F, still on the title of y axis report the definition and then (%)
There might be a misunderstanding. In Figure 2A, 2C, 2E the intracellular growth coefficient (IGC), defined as IGC= (CFU6h – CFUinvasion)/ CFUinvasion is depictured. The IGC has no unit. We now changed into IGC in Caco2, ect.
In Figure 2B, 2D, 2F the % of GFP-positive cells, a measurement for cell-to-cell spread is depicted. We now changed into % of GFP-positive Caco2 cells, etc.
Additionally, there wasn’t any statistically significant difference between the IGC of the four strains (p<0.05).
Discussion is professionally written, although some considerations on the potential of this study should be included. What this study could give, which is the novel contribution on knowledge that this paper could fill? To whom is targeted?
Introduction was modified. A passage was introduced to the Discussion section.
References section contains many typos:
to >i>Listeria monocytogenes
Nat. Rev. Immunol. 2004.
Appl Env. Microbiol
We have corrected the references.
Reviewer 3 Report
This is a well written and very clear manuscript. The results are useful and interesting to a broad microbiological audience and will progress the field. Some very minor english typos remain in what is otherwise a manuscript ready for publication.
Author Response
This is a well written and very clear manuscript. The results are useful and interesting to a broad microbiological audience and will progress the field. Some very minor english typos remain in what is otherwise a manuscript ready for publication.
We have carefully revised the manuscript. The manuscript was additionally revised by a native speaker.
Reviewer 4 Report
Thank you for giving me the oprortunity to review your article. I believe thr Introductiin part and the discussion section may be improved.
Author Response
Thank you for giving me the oprortunity to review your article. I believe thr Introductiin part and the discussion section may be improved.
We revised the introduction including more information regarding the infection pathway and the major virulence factors. Furthermore, we included a conclusion and in the discussion section.
Round 2
Reviewer 1 Report
The revised version of the manuscript shows clear improvements. Most comments are in the attached file.
I'd encourage the authors to include in the text, where pertinent, what stated in the last two responses to my comments, in order to make more explicit their choice of referring only to virulence genes that showed different presence/absence patterns, different length compared to the described references, or lower (i.e., <90%) amino acid identity.
Also the perspective of "Additional experiments e.g. transcriptome analysis of bacteria isolated from organs, are needed to determine the molecular mechanism behind the long-term survival of L. monocytogenes strains in organs." might be worthy of beeing included in the discussion.
Author Response
Second Revision 10_08_2021
I'd encourage the authors to include in the text, where pertinent, what stated in the last two responses to my comments, in order to make more explicit their choice of referring only to virulence genes that showed different presence/absence patterns, different length compared to the described references, or lower (i.e., <90%) amino acid identity.
A text passage reflecting on this consideration was included.
Also the perspective of "Additional experiments e.g. transcriptome analysis of bacteria isolated from organs, are needed to determine the molecular mechanism behind the long-term survival of L. monocytogenes strains in organs." might be worthy of beeing included in the discussion.
Excellent idea. This sentence was included.